# Vitamin D May Protect against Breast Cancer through the Regulation of Long Noncoding RNAs by VDR Signaling

**DOI:** 10.3390/ijms23063189

**Published:** 2022-03-16

**Authors:** Janusz Blasiak, Jan Chojnacki, Elzbieta Pawlowska, Aleksandra Jablkowska, Cezary Chojnacki

**Affiliations:** 1Department of Molecular Genetics, Faculty of Biology and Environmental Protection, University of Lodz, 90-236 Lodz, Poland; 2Department of Clinical Nutrition and Gastroenterological Diagnostics, Medical University of Lodz, 90-647 Lodz, Poland; jan.chojnacki@umed.lodz.pl (J.C.); cezary.chojnacki@umed.lodz.pl (C.C.); 3Department of Orthodontics, Medical University of Lodz, 92-217 Lodz, Poland; elzbieta.pawlowska@umed.lodz.pl; 4Department of Internal Diseases and Cardiology, H. Jonscher Hospital, 93-113 Lodz, Poland; jablkowska@pro.onet.pl

**Keywords:** breast cancer, vitamin D3, 1,25(OH)2D, VDR, lncRNA, miRNA

## Abstract

Dietary vitamin D3 has attracted wide interest as a natural compound for breast cancer prevention and therapy, supported by in vitro and animal studies. The exact mechanism of such action of vitamin D3 is unknown and may include several independent or partly dependent pathways. The active metabolite of vitamin D3, 1α,25-dihydroxyvitamin D3 (1,25(OH)2D, calcitriol), binds to the vitamin D receptor (VDR) and induces its translocation to the nucleus, where it transactivates a myriad of genes. Vitamin D3 is involved in the maintenance of a normal epigenetic profile whose disturbance may contribute to breast cancer. In general, the protective effect of vitamin D3 against breast cancer is underlined by inhibition of proliferation and migration, stimulation of differentiation and apoptosis, and inhibition of epithelial/mesenchymal transition in breast cells. Vitamin D3 may also inhibit the transformation of normal mammary progenitors into breast cancer stem cells that initiate and sustain the growth of breast tumors. As long noncoding RNAs (lncRNAs) play an important role in breast cancer pathogenesis, and the specific mechanisms underlying this role are poorly understood, we provided several arguments that vitamin D3/VDR may induce protective effects in breast cancer through modulation of lncRNAs that are important for breast cancer pathogenesis. The main lncRNAs candidates to mediate the protective effect of vitamin D3 in breast cancer are *lncBCAS1-4_1,* AFAP1 antisense RNA 1 (*AFAP1-AS1*), metastasis-associated lung adenocarcinoma transcript 1 (*MALAT1*), long intergenic non-protein-coding RNA 511 (*LINC00511*), *LINC00346*, small nucleolar RNA host gene 6 (*SNHG6*), and *SNHG16*, but there is a rationale to explore several other lncRNAs.

## 1. Introduction

Despite significant recent progress in diagnosis and therapy, breast cancer is still a serious problem for individuals and societies (reviewed in [1]). There are several reasons for this, including heterogeneity of cases and poorly understood mechanisms of breast cancer pathogenesis. Although surgery, chemo- and radiotherapy, and hormone therapy are still the basic modes of breast cancer treatment, several chemical agents have been considered to assist breast cancer therapy and/or recommended in breast cancer prevention (reviewed in [2]). Some substances that can be administered with the diet, including vitamins, have been reported to improve the treatment of or play a preventive role in breast cancer ([3,4,5,6]). Several reports have presented vitamin D to have a beneficial potential in breast cancer (e.g., [7,8,9,10]).

Vitamin D3 (cholecalciferol), which is derived from animals, is the main type of vitamin D, and its biologically active form, 1α,25-dihydroxyvitamin D3 (1,25(OH)2D, calcitriol), is also an endocrine hormone. The serum concentration of the stable precursor of 1,25(OH)2D, 25-hydroxyvitamin D (25(OH)D, calcidiol) is usually used as a marker of vitamin D3 concentration in an organism. The biological activity of 1,25(OH)2D is underlined by its binding to the vitamin D3 receptor (VDR), a member of the nuclear receptor superfamily of ligand-inducible transcription factors. This 1,25(OH)2D/VDR complex is often associated with retinoid X receptor alpha (RXRA), and together, they bind the vitamin D response elements (VDREs) in the promoters of many genes to activate or repress their transcription [11]. Downstream targets of VDR are a myriad of genes, including those involved in genome maintenance, immune response, and cancer (reviewed in [12,13]).

Although data from epidemiological studies associating the levels of 25(OH)D with a better outcome of breast cancer are not fully consistent, most imply a positive correlation. A meta-analysis suggested that high vitamin D3 levels are weakly associated with a low breast cancer risk but strongly associated with the survival of breast cancer patients [14]. A more recent meta-analysis confirmed that lower blood 25(OH)D levels are associated with decreased survival in breast cancer patients [15]. Experimental studies have provided more convincing arguments for the positive role of vitamin D3 in breast cancer prevention and treatment. The BRCA1 DNA-repair-associated, breast cancer type 1 susceptibility (*BRCA1*) gene, which is important in the pathogenesis of a subset of breast cancer cases (reviewed in [16]), was transactivated by VDR-induced factors [17]. This study also showed the BRCA1-dependent antiproliferative action of 1,25(OH)2D in breast cancer cell lines. Several other studies have shown the important role of the vitamin D3/VDR axis in breast cancer pathogenesis (reviewed in [8]). The general conclusion from these studies is that VDR may act as a tumor suppressor whose regulation is impaired in breast cancer transformation [18]. However, the mechanism underlying this beneficial potential of vitamin D and its receptor is unknown.

Vitamin D metabolism and signaling in cancer are determined by various factors, including regulatory noncoding RNAs (ncRNAs) (reviewed in [19]). On the other hand, these RNAs can be regulated by vitamin D3/VDR signaling. Although most studies on these mutual interactions have focused on microRNAs (miRNAs), the role of long noncoding RNAs (lncRNAs) is emerging [19].

The basic functions of lncRNAs in the regulation of gene expression can be arbitrarily divided into antisense/signaling, guide, scaffold, and decoy. A lncRNA may pair with a complementary fragment of mRNA, preventing or inhibiting its translation. Transcription can be influenced by a lncRNA through recruitment and/or by guiding transcriptional activators and repressors to the target gene. A lncRNA may serve as a platform (scaffold) to facilitate the assembly of a chromatin remodeling complex to alter the structure of chromatin to be more permissive or repressive for transcription factors. A lncRNA may act as a decoy to recruit other regulatory RNAs or transcription factors and sequester them from their target mRNA or DNA. The “sponging” effect occurs when a single lncRNA binds many miRNAs, preventing interaction with their targets. Several lncRNAs have been identified to modulate VDR signaling in breast cancer [20,21].

Therefore, lncRNAs can be involved in vitamin D3/VDR signaling, and the vitamin may regulate the expression of lncRNAs that may be important in breast cancer. This regulatory circuit justifies studies on the role of lncRNAs as a mediator of the effects of vitamin D in breast cancer. This manuscript summarizes and updates information on the effects of vitamin D3/VDR in breast cancer, the mutual interaction between vitamin D3/VDR and lncRNAs, and the role lncRNAs may play in the effects of vitamin D in breast cancer pathogenesis. In our previous work, we reviewed the role of vitamin D in triple-negative and BRCA1-deficient breast cancer cases [8].

In this review, the term “vitamin D3” is used in a broad sense. In in vivo studies, either chemically pure vitamin D3 or its modifications are used, and the concentration of 25(OH)D in serum is determined. In in vitro studies, typically lacking a vitamin-D3-metabolizing system, 1,25(OH)2D is used. This is why generalizing 1,25(OH)2D-related results to vitamin D3 may be chemically incorrect but, usually, biologically justified. For a concise review, we use “vitamin D3” in a general sense, as the term vitamin D receptor is commonly used despite being the 1,25(OH)2D receptor.

## 2. Vitamin D Signaling in Breast Cancer

Synthesis of vitamin D3 in the skin requires UV radiation, which can induce DNA damage, preferentially cyclobutane pyrimidine dimers and (6-4)-pyrimidine-pyrimidone photoproducts, which are found in many skin cancers, including melanoma (reviewed in [22]). On the other hand, vitamin D3 protects the skin against UV-induced aging and DNA damage causing cancer. This raises the question about the general anticancer properties of vitamin D3.

Usually, the biological effects of vitamin D are classified into nongenomic and genomic (reviewed in [23]). Genomic effects are mediated by VDR, RXR, and VDREs and result in long-term, sometimes delayed, biological consequences. The nongenomic pathway includes rapid cellular effects of 1,25(OH)2D, such as the protection of VDR-defective human fibroblasts against DNA damage induced by UV mediated by endoplasmic reticulum stress protein 57 (ERP57, MARRS) [24]. Therefore, the anticancer effects of vitamin D3 may be underlined by its genomic and nongenomic actions (reviewed in [25]).

The anticancer effect of vitamin D3 is not limited to UV-induced skin cancers. Colston et al. and Abe et al. were the first to show the anticancer properties of vitamin D in vitro [26,27]. Subsequently, many in vivo and in vitro studies showed the protective action of vitamin D against various cancers and identified several genes important in cancer transformation as targets in the genomic action of vitamin D (reviewed in [9]). In general, the antitumor effects of vitamin D are underlined by modulation of specific signaling pathways that are involved in tumor growth (Figure 1). Tumor growth is determined by the fate of cancer cells that may differentiate, arrest the cell cycle, and undergo apoptosis and/or degradation.

Vitamin D3/VDR signaling is implicated in the morphogenesis of the postnatal mammary gland, negatively regulating its growth [28]. Therefore, impaired vitamin D/VDR signaling may promote the abnormal development of the mammary gland, which may contribute to young age-onset breast cancer cases.

Malignant breast tumors are formed and maintained with the involvement of breast cancer stem cells (BCSCs) [29]. Moreover, BCSCs are involved in progression to metastasis, relapse, and therapy resistance. 1,25(OH)2D and a gemini analog of vitamin D3 (BXL0124) were reported to inhibit mammosphere formation from mammary progenitors and downregulated the pluripotency markers octamer-binding transcription factor 4 (OCT4) and Kruper like factor 4 (KLF-4) in mammospheres [30]. 1,25(OH)2D also repressed markers of stem cell-like phenotypes, including cluster of differentiation 44 (CD44), CD49f, cleaved Notch receptor 1 (c-Notch1), and phosphorylated nuclear factor kappa-light-chain-enhancer of activated B cells (pNFκB) in breast cancer. Subsequently, it was shown that 1,25(OH)2D and BXL0124 downregulated the pluripotency markers OCT4, CD44, and laminin subunit alpha 5 (LAMA5) in the mammospheres of a triple-negative breast cancer (TNBC) cell line [31]. Furthermore, 1,25(OH)2D and its analog repressed other proteins important for the maintenance of BCSCs, including Notch1-3, jagged canonical Notch ligand 1 (JAG1-2), Hes family BHLH transcription factor 1 (HES1), and NFκB. Therefore, vitamin D3/VDR signaling may prevent acquiring a stem-like phenotype by normal mammary gland progenitors and somatic breast cancer cells (Figure 2). Several pathways involved in cancer stemness are modulated by lncRNAs, and their targeting may be considered a new strategy to control tumor development, invasion, metastasis, and therapeutic resistance (e.g., [32,33,34]).

The hormonal properties of vitamin D3, as part of the vitamin D3 endocrine system, may be responsible for the increased efficacy of tamoxifen, a selective inhibitor of estrogen receptor (ER) commonly applied in breast cancer therapy [35]. Such an action of vitamin D seems to be especially important in the case of breast cancer in postmenopausal women who no longer maintain ovarian production of estrogen [36]. Aromatase is a key enzyme in estrogen synthesis, and vitamin D was reported to inhibit it and repress the expression of estrogen receptor alpha (ERα) in breast tissue, [37,38,39,40,41]. Therefore, vitamin D is particularly important in breast cancer ER-positive cases in postmenopausal women. Many lncRNAs were reported to play a role in the regulation of estrogen-dependent signaling (reviewed in [42]).

The role of autophagy in cancer in general and in breast cancer in particular is emerging [43,44,45]. Regulation of autophagy is considered a method to prevent breast cancer and break therapeutic resistance (reviewed in [46]). It was shown that the vitamin D/VDR axis in breast cancer is regulated by many miRNAs [47]. Upregulation of *miR-214* diminished VDR-mediated signaling in breast cancer cell lines with a concomitant positive correlation between VDR level and an inhibitor of the Hedgehog pathway [48]. The *VDR* gene has a sequence in its 3′-untranslated region recognized by *miR-125b* whose overexpression in the MCF-7-line decreases VDR level [49]. On the other hand, 1,25(OH)2D inhibited the expression of *miR-125b* in MCF-7 cells [50]. These results highlight the role of lncRNAs in mediating effects of vitamin D3 in breast cancer as one of the main mechanisms of biological action of lncRNAs is miRNAs sponging.

Other pathways and mechanisms may contribute to vitamin D3/VDR signaling in breast cancer (reviewed in [10]). The results of research on breast cancer cell lines should be interpreted considering the high heterogeneity of breast cancer cases and different forms of VDR, as well as pleiotropy of biological action of vitamin D [51]. Breast cancer cells, depending on origin, can be inherently vitamin D3 resistant or sensitive or acquire resistance during disease progression due to various effects, including methylation of the *VDR* promoter [52]. Heterogeneity of breast cancer also involves differences between specific breast compartments, which cause differential distribution of vitamin D3 in different regions of breast tumors due to different activity of enzymes involved in vitamin D3/VDR signaling.

In summary, several epidemiological studies show vitamin D3 deficiency in breast cancer patients. These association studies are supported by experimental research, but the mechanism underlying the potential beneficiary effect of vitamin D3 in breast cancer remains poorly understood. To apply vitamin D3 in breast cancer prevention and therapy effectively and safely, more controlled clinical trials and mechanistic laboratory studies are needed. 1,25(OH)2D may directly interact with several membrane-bound proteins to initiate a cascade of several signaling pathways that control cell growth and cell cycle arrest, cell migration, proliferation, and apoptosis important for cancer transformation.

## 3. Long Noncoding RNAs in the Vitamin D Signaling in Cancer

As presented in Section 2, many miRNAs may be implicated in the regulation of the vitamin D3/VDR axis in breast cancer. The miRNA/mRNA/lncRNA interaction and direct binding of miRNAs by a lncRNA (a “sponge” effect) are the main mechanisms beyond the effect of lncRNAs on gene expression [53]. The role of lncRNAs in cancer in general and in breast cancer in particular is an emerging field of study, especially from the perspective of the Cancer lncRNA Census, but it is not the subject of this review, except in the context of vitamin D3 signaling. This topic is addressed in many excellent reviews, e.g., [54,55].

Abnormal differentiation is a hallmark of cancer transformation [56]. Hou et al. showed that (+)-cholesten-3-one (CN)-induced differentiation of rat bone marrow stromal cells (MSCs) into osteoblasts required VDR activation [57]. They then identified almost 300 lncRNAs whose expression was changed during MSC differentiation, but a direct association between the expression of these lncRNAs and vitamin D3/VDR signaling was not found [58].

1,25(OH)2D was shown to suppress cyclins D1 and Gli1, which are regulated by the β-catenin/TCF signaling pathway and play an essential role in epidermal carcinogenesis [59]. Suppression of VDR in mice evoked hyperproliferation of keratinocytes and increased expression of cyclins D1 and Gli1.

Jiang et al. profiled lncRNAs from mouse keratinocytes cultured in vitro and mouse epidermis from epidermal-specific *VDR*-null mice and their normal littermates [59]. They found that several lncRNAs with annotated functions of oncogenes, including H19 imprinted maternally expressed transcript (*H19*), HOXA distal transcript antisense RNA (*HOTTIP*), and GNAS antisense RNA 1 (*Nespas*) were upregulated, whereas tumor-suppressing lncRNAs, including KCNQ1 opposite strand/antisense transcript 1 (*Kcnq1ot1*) and *lincRNA-p21*, were downregulated in these cells. An analogous array of lncRNA expression was observed in the epidermis of epidermal-specific *VDR*-null mice compared with control littermates. This study revealed a novel mechanism of anticancer action of vitamin D/VDR through maintenance of the right proportion of oncogenic to tumor-suppressing lncRNAs.

Zuo et al. found that upregulation of the maternally expressed gene 3 (*MEG3*) lncRNA inhibited glycolysis, glycolytic capacity, and lactate production in primary colorectal cancer (CRC) cells and CRC cell lines, but downregulation of *MEG3* caused a reverse effect [60]. Overexpression of *MEG3* induced ubiquitin-dependent degradation of BHLH transcription factor, MYC proto-oncogene (c-Myc), and inhibited its downstream genes of the glycolysis pathway. Furthermore, vitamin D3 and VDR activated *MEG3*. *MEG3* expression was positively correlated with 25(OH)D concentration in the serum of CRC patients, 1,25(OH)2D increased *MEG3* expression, and knockdown of the *VDR* gene abolished the effect of *MEG3* on glycolysis. These results suggest that vitamin-D3-activated *MEG3* inhibits aerobic glycolysis in CRC cells through degradation of c-Myc. A protective effect of vitamin D3 against endothelial cell damage induced by high glucose concentrations was observed [61]. This effect was mediated by increased expression of *MEG3* and depression of toll-like receptor 4 (TLR4)/myeloid differentiation factor 88 (MyD88)/NF-κB(p65) signaling. Downregulation of *MEG3* reduced the protective effect of vitamin D3 on diabetes-related cell damage.

High-throughput data analysis to identify lncRNAs involved in ovarian cancer showed that high expression of the TOPORS antisense RNA 1 (*TOPORS-AS1*) lncRNA was associated with higher overall survival in ovarian cancer patients compared with patients exhibiting lower *TOPORS-AS1* expression [62]. These clinical studies were supported by observations that overexpressing *TOPORS-AS1* in ovarian cancer cells suppressed cell proliferation, migration, invasion, and colony formation. Associated transcriptomic analysis showed the involvement of *TOPORS-AS1* in Wnt/β-catenin signaling. Moreover, *TOPORS-AS1* increased the phosphorylation of β-catenin and suppressed the expression of catenin beta 1 (*CTNNB1*), impairing the Wnt/β-catenin pathway. These studies also showed that VDR upregulated *TOPORS-AS1*, and inhibition of β-catenin by *TOPORS-AS1* required heterogeneous nuclear ribonucleoprotein A2B1 (hnRNPA2B1), an RNA-binding protein. Therefore, VDR may stimulate *TOPORS-AS1* to exert its action as a tumor suppressor in ovarian cancer through disruption of Wnt/β-catenin signaling.

Xue et al. found over 600 lncRNAs with expression enriched in ovarian cancer cells in several pathways, including vitamin D signaling [63]. They found that the lncRNA *lncBCAS1-4_1* linked vitamin D signaling with epithelial/mesenchymal transition (EMT), an important effect in cancer invasion and metastasis. Moreover, *lncBCAS1-4_1* showed mostly the same transcripts with CYP24A1, an essential enzyme in 1,25(OH)2D metabolism. Knockdown of *lncBCAS1-4_1* suppressed the proliferation and migration of ovarian cancer cells. Therefore, *lncBCAS1-4_1* is an important element in the anticancer mechanism of 1,25(OH)2D.

In summary, various mechanisms may be involved in the anticancer action of vitamin D and its receptor mediated by lncRNAs (Figure 3). These mechanisms are associated with the general properties of lncRNAs as oncogenes or tumor suppressors, as well as their role in the regulation of signaling pathways important for cancer transformation.

## 4. Long Noncoding RNAs and Vitamin D Signaling in Breast Cancer

Oskooei et al. found upregulation of the *VDR* gene, as well as *MALAT1* and *LINC00511* lncRNAs, in breast cancer samples from 75 patients compared to their adjacent noncancerous tissues [20]. These authors also observed positive correlations between the expression level of the small nucleolar RNA host gene 16 (*SNHG16*) and *LINC00511* genes and nuclear grade, tubule formation, and family history of cancer. In addition, a correlation between the expression of *VDR* and progesterone receptor status was observed. Furthermore, an association between the expression of *VDR* and *SNGH16* was reported in both tumor and nontumor tissues.

Xi et al. identified the AFAP1 antisense RNA 1 (*AFAP1-AS1*) lncRNA as a TNBC-specific gene linked with cell proliferation and EMT [64]. These studies also showed increased activity of the vitamin D3 biosynthesis pathway. Although a direct correlation between lncRNA expression and vitamin D3/VDR signaling in breast cancer was not shown in this study, the results suggest the possibility that *AFAP1-AS1* may regulate genes that are important for this signaling, as many genes involved in vitamin D3 metabolism were identified as active. Some lncRNAs that were reported to play a role in the effect of vitamin D3 in breast cancer are presented in Table 1.

LncRNAs expression has been associated with the occurrence and progression of breast cancer in many studies. On the other hand, these lncRNAs may be involved in vitamin D3 signaling. Oskooei and Ghafouri-Fard presented the results of a literature search to support their hypothesis that the expression of some lncRNAs positively correlates with the expression of estrogen receptor 1 (ESR1) and negatively correlates with *VDR* expression [72]. They also applied in silico analysis to identify lncRNAs that could oppositely affect these two pathways. They also assessed the interaction between these lncRNAs and some miRNAs that may act as oncogenes/tumor suppressor genes.

GATA-binding protein 3 (GATA3) is a zinc-finger-type transcription factor that recognizes the AGATAG motif in the target gene, whose genetic constitution is important in breast cancer [73]. The *GATA3-AS1* lncRNA belongs to important regulators of GATA3 expression and has been reported to correlate with clinical/pathological features of breast cancer [74,75]. In turn, vitamin D3 was reported to influence GATA3 expression [76]. Moreover, *GATA3-AS1* was reported to interact with some oncogenes and tumor suppressor genes, such as zinc-finger protein 217 (*ZNF217*)*,* phosphatase and tensin homolog (*PTEN, TP53 tumor protein p53*), and RB transcriptional corepressor (*RB1*), which predispose it to play an important role in the pathogenesis of many cancers [72]. Collectively, these data suggest an involvement of the *GATA3-AS1* lncRNA in breast cancer pathogenesis through vitamin D3/VDR signaling.

Small nucleolar RNA host gene 12 (*SNHG12*), a transcriptional regulator of *c-Myc*, interacts with other genes important for cancer transformation, including oncogenes and tumor suppressor genes, such as *TP53*, *PTEN*, and titin *(TTN*) [77]. *SNHG12* was reported to increase in breast cancer tissues and cells and correlate with cancer progression [78,79]. C-Myc-induced upregulation of *SNHG12* was shown to support breast cancer cell proliferation and migration [79]. Subsequently, it was shown that these cancer-related effects of *SNHG12* are underlined by sponging *miR-451a* [78].

As mentioned previously, the *H19* lncRNA was suggested to be involved in the maintenance of the right proportion of oncogenic to tumor-suppressing lncRNAs in a vitamin D3/VDR signaling-dependent manner [80]. However, *H19* was upregulated in breast cancer tissue and cell lines and antagonized TP53/p53 and its target gene, TNF alpha-induced protein 8 (*TNAIP8*). Consequently, it impaired the expression of EMT genes [81]. Moreover, *H19* is considered to contribute to breast cancer pathogenesis by several other mechanisms, including the regulation of *miR-675* and interaction with c-Myc [81,82]. Consequently, *H19* is considered a breast cancer diagnostic and prognostic marker [83,84].

*HOTTIP*, in addition to its regulation of the right proportion between oncogenic and tumor-suppressing lncRNAs, was reported to promote migration, invasiveness, and EMT of breast cancer cells by regulating the Wnt/β-catenin pathway [85]. A subsequent study showed that *HOTTIP* increased the expression of pluripotency markers octamer-binding transcription factor 4 (*OCT4*) and SRY-box transcription factor 2 (*SOX2*), and, in this way, it facilitated the stemness of breast cancer cells by regulating the *miR-148a-3p*/Wnt1 pathway [86].

The lncRNA *Nespas* has an annotated function as an oncogene and is regulated by VDR signaling [80]. It encodes *miR-296-5p*, which was reported to target the catalytic subunit of telomerase in breast cancer cells [87]. Ectopic *miR-296-5p* expression reduced telomerase activity, drove telomere shortening, and caused proliferation defects by enhancing senescence and apoptosis in breast cancer cell lines, but the expression of *Nespas* and *miR-296-5p* was reduced in human basal-type breast cancer tissue, contributing to the higher aggressiveness of this type of the breast cancer.

*Kcnq1ot1* is another lncRNA regulated by VDR signaling [80]. It was also shown to play an important role in breast cancer transformation [88,89]. It was concluded that *Kcnq1ot1* is essential for the development of the epithelium of the normal human mammary gland, although this conclusion was drawn from the results obtained in the MCF-7 breast cancer cell line [89]. *Kcnq1ot1* was upregulated and modulated cyclin E2 (CCNE2) in breast cancer tissues and cells through sponging *miR-145* [88]. This study clearly indicated *Kcnq1ot1* as an oncogene. This conclusion was confirmed in subsequent studies showing upregulation of *Kcnq1ot1* in breast cancer cell lines [90]. *Kcnq1ot1* knockdown compromised cancer cell functions, including proliferation and migration, as well as in vivo transplant growth. The possible mechanism underlying these effects could be sponging and regulation of *has-miR-107*.

*LincRNA-p21*, also called p53 pathway corepressor 1 protein tumor (*TRP53COR1*), is associated with VDR signaling [80]. It was shown that its expression was induced by chemotherapy and mediated by a mutated p53 form that targeted the G quadruplex structure rather than the p53 response elements in its promoter [91]. It was shown that *lincRNA-p21* was upregulated in 4T1 breast tumor-associated macrophages (TAMs) [92]. TAMs with *lincRNA-p21* knockdown evoked cancer cell apoptosis and suppressed tumor cell migration and invasion by promoting MDM2 to antagonize p53 activation. This suggests that *lincRNA-p21* may play an indirect role in the oncogene as a major regulator of TAM function in the breast tumor environment.

The *MEG3* lncRNA was reported to be activated by vitamin D3 and VDR [60]. This lncRNA controls the expression of many tumor suppressor genes, including *p53*, *RB1*, *c-Myc*, and transforming growth factor-beta *(TGFβ*) [93]. It affects EMT through modulation of Wnt/β-catenin signaling. It was reported that *MEG3* overexpression caused inhibition of proliferation and invasion of breast cancer cells [94]. Upregulated *MEG3* also suppressed capillary tube formation in endothelial cells and angiogenesis in vivo and was underlined by downregulation of AKT serine/threonine kinase (Akt) signaling. Bioinformatic analysis showed that estrogen receptor and progesterone receptor status in breast cancer were positively correlated with *MEG3* expression [95]. Moreover, basal-like status, TNBC, and Bloom/Richardson status were negatively correlated with *MEG3* expression. Furthermore, there was a positive correlation between *MEG3* and heparan sulfate proteoglycan 2 expression in breast cancer tissues [95].

HOX transcript antisense RNA (*HOTAIR*) is a lncRNA with an emerging role in breast cancer pathogenesis (reviewed in [96]). Its putative connection with vitamin D3/VDR signaling was shown in a multiple sclerosis study [97]. The main role of *HOTAIR* in breast cancer pathogenesis lies in the promotion of metastasis, and many pathways may lie behind this role, including those presented previously (Figure 1): TGFβ, Wnt/β-catenin, PI3K/Akt/MAPK, and vascular endothelial growth factor (VEGF) [98]. This makes *HOTAIR* a candidate prognostic marker in breast cancer [99,100]. It was shown that *HOTAIR* was upregulated in breast cancer tissues, and its knockdown inhibited the proliferation, migration, invasion, and activity of the Akt signaling of breast cancer cells [101]. In addition, *HOTAIR* served as a sponge for *miR-601*, and a *miR-601* inhibitor reversed the effect of *HOTAIR* silencing on breast cancer progression. Zinc-finger E-box-binding homeobox 1 (*ZEB1*) was targeted by *miR-601* and directly regulated by *HOTAIR*. Therefore, HOTAIR may regulate breast cancer progression by controlling the *miR-601/ZEB1* axis.

As mentioned previously, *TOPORS-AS1* may be stimulated by VDR to exert its action as a tumor suppressor in ovarian cancer through disruption of Wnt/β-catenin signaling [62]. An early breast cancer study also confirmed the tumor-suppressing role of *TOPORS-AS1* [102]. However, subsequent research showed an association of the high expression of *TOPORS-AS1* with a good/poor outcome in ER-positive/negative patients [103]. Therefore, the association of *TOPORS-AF1* expression with hormone status in breast cancer needs further research. The mechanism of *TOPORS-AF1* action in breast cancer is not fully understood, but according to the origin of this lncRNA from the topoisomerase I binding, arginine/serine-rich, E3 ubiquitin-protein ligase (TOPORS), it is considered to exert its tumor-suppressing action through regulation of breast cancer cell proliferation and apoptosis and regulation of p53/TP53 stability via ubiquitin-dependent degradation [102].

Some results on the putative involvement of lncRNAs in mediating vitamin D3 effects in breast cancer are summarized in Table 2.

## 5. Conclusions and Perspectives

Breast cancer is still a serious challenge for clinicians and scientists due to incomplete knowledge on its pathogenesis and great heterogeneity of cases. The use of vitamin D3 in breast cancer prevention and therapy has been considered for years, but a solid conclusion on this subject has not been drawn. However, it should be stressed that vitamin D3 is considered to a much greater extent as a preventive than a therapeutic compound.

Breast cancer pathogenesis involves many signaling pathways that include the expression of many genes controlled at genetic and epigenetic levels. lncRNAs constitute about 80% of the entire human transcriptome, and so it has a high potential to regulate gene expression, especially since a single lncRNA may modulate the action of many miRNAs, whose regulatory role is well established [103]. Therefore, lncRNAs may be involved in the regulation of both normal development and pathogenesis of many, if not all, diseases. The problem is to identify these lncRNA species that may be targeted in the prevention and/or therapy of a specific disease.

Many lncRNAs have been identified to be implicated in breast cancer pathogenesis. On the other hand, several lncRNAs are important for vitamin D3/VDR signaling. Therefore, these lncRNA molecules play a role in both effects and are candidates contributing to the protective effect of vitamin D3 in breast cancer. To date, many such lncRNAs have not been identified so far, e.g., *AFAP1-AS1, MALAT1, LINC00511, LINC00346, SNHG6*, and *SNHG16*. However, there are rationales to further explore other lncRNAs, including *GATA3-AS1, H19, HOTAIR, HOTTIP, Kcnq1ot1, lincRNA-p21, MEG3, Nespas, SNHG12,* and *TOPORS-AS1*. Most of these lncRNAs play an established role in cancer transformation as oncogenes or tumor suppressor genes. Their further exploration in the context of vitamin D3/VDR signaling may provide details on the effects of vitamin D3 in breast cancer mediated by lncRNAs.

Exploring the role of lncRNAs on the effects of vitamin D3 in breast cancer may be especially difficult due to (1) heterogeneity of breast cancer cases, (2) pleiotropic biological effects of vitamin D3, and (3) a great number of genes regulated by almost all, if not all, lncRNAs [51]. Therefore, special attention should be paid to the determination of specific breast cancer cases in clinical research and the homogeneity of breast cancer cell lines in laboratory studies. This may limit the problem with the different effects of vitamin D3 in different breast cancer types. It can be expected that vitamin D3 may be effective in some specific breast cancer types, whereas its efficiency in others can be low, if at all, and analysis of lncRNAs involved in vitamin D3 action may help predict the output of such action. The results of studies on the cancer-related nature of lncRNAs, oncogenic or tumor suppressing, may also help to foresee the effect of vitamin D3 as either beneficial or harmful.

## Figures and Tables

**Figure 1 ijms-23-03189-f001:**
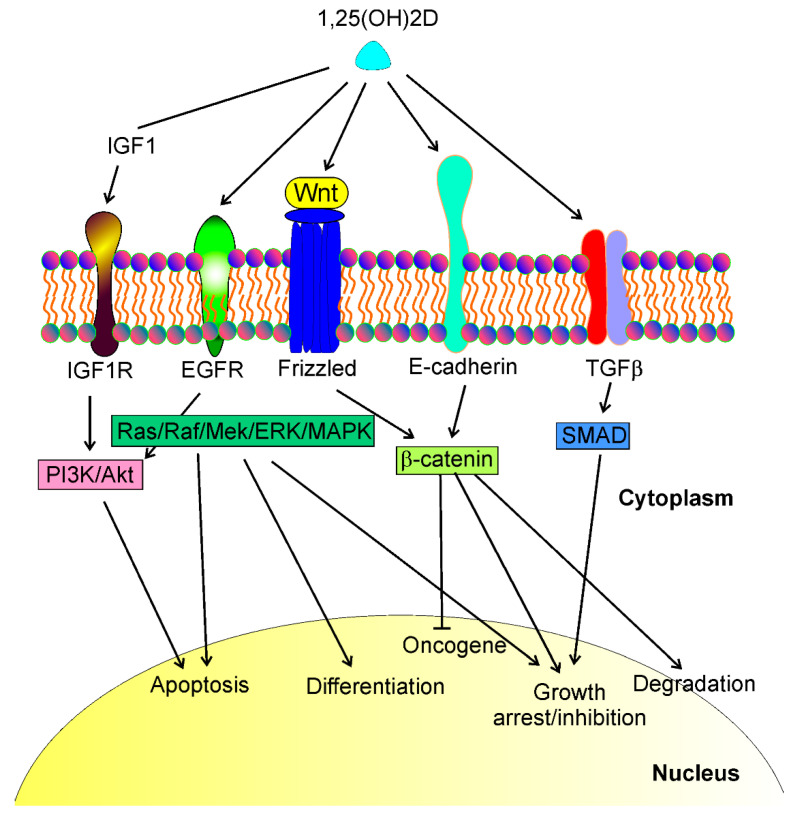
1,25(OH)2D, a biologically active metabolite of vitamin D, may interact with different transmembrane proteins and affect several signaling pathways that regulate essential phenomena in cancer cells such as apoptosis, differentiation, growth arrest, cell cycle, oncogene and tumor suppressor expression, degradation. These proteins include, but are not limited to, insulin growth factor 1 (IGF1) receptor (IGF1R), epidermal growth factor receptor (EGFR), Wnt/frizzled proteins, E-cadherin, and transforming growth factor-beta (TGFβ). Interactions of 1,25(OH)2D with these proteins lead to a cascade of downstream events resulting in the activation of several signaling pathways, including PI3K/Akt, Ras/Raf/Mek/ERK/MAPK, β-catenin, and SMAD (phosphatidylinositol 3-kinase/AKT serine/threonine kinase, KRAS proto-oncogene/Raf-1 proto-oncogene/mitogen-activated protein kinase kinase 1, SMAD family members, respectively). These signaling pathways may directly regulate the life and death of cancer cells or control the transcription of genes whose products contribute to cancer transformation.

**Figure 2 ijms-23-03189-f002:**
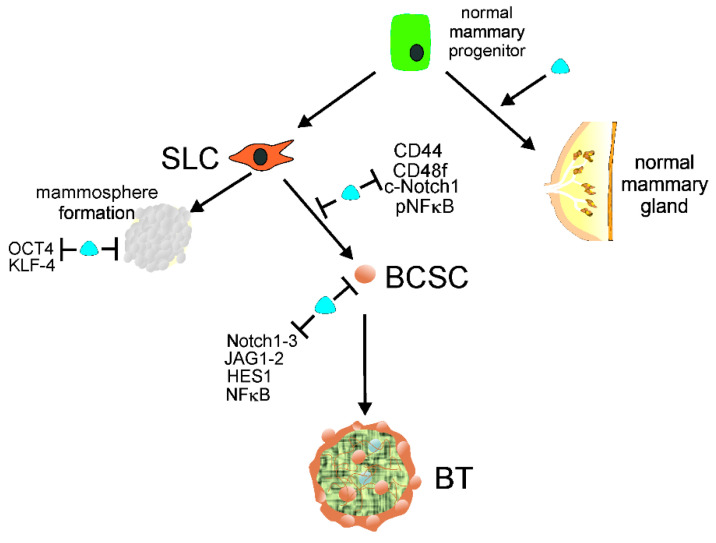
Normal mammary gland progenitors may develop into a normal mammary gland or acquire stem cell-like (SLC) properties to form mammospheres or/and transform into breast cancer stem cells (BCSCs). 1,25(OH)2D, a biologically active metabolite of vitamin D, along with its receptor, VDR, symbolized here as a blue tear, supports the normal development of mammary gland and prevents mammosphere formation by repressing the pluripotency markers OCT4 and KLF-4. Vitamin D may also prevent the conversion of SLC cells into BCSCs by downregulation of other pluripotency markers, including CD44, CD49f, c-Notch1, and pNFκB. Vitamin D may also repress proteins needed for the maintenance of BCSCs, such as Notch1-3, JAG1-2, HES1, and NFκB, preventing the initiation and formation of breast tumors (BTs). Abbreviations are defined in the main text.

**Figure 3 ijms-23-03189-f003:**
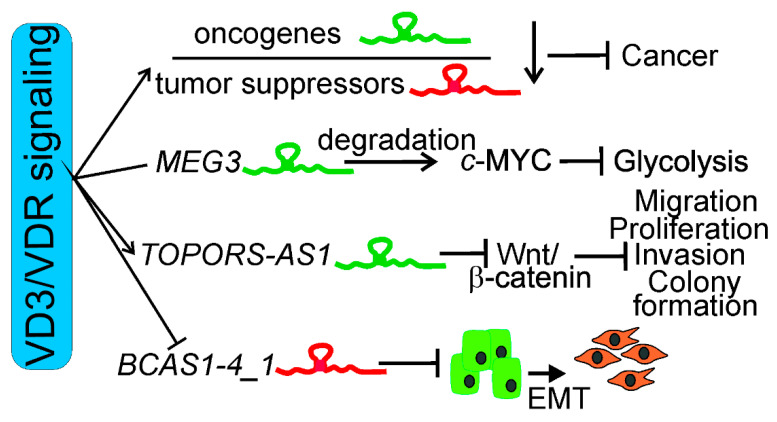
Different mechanisms of anticancer action of vitamin D3 (VD3) and its receptor, VDR, supported by long noncoding RNAs (lncRNAs). Vitamin D3 and VDR may decrease the expression of oncogenic lncRNAs and/or increase the expression of their tumor-suppressing counterparts. Vitamin D may activate lncRNAs that degrade BHLH transcription factor, MYC proto-oncogene (c-Myc), causing inhibition of glycolysis in cancer cells, as shown for the maternally expressed gene 3 (*MEG3*) lncRNA and the colorectal cancer cell line. Vitamin D3/VDR may stimulate a tumor-suppressing lncRNA to disrupt Wnt/β-catenin signaling, resulting in inhibition of migration, proliferation, invasion, and colony formation of cancer cells, as shown for TOPORS antisense RNA 1 (*TOPORS-AS1*). 1,25(OH)2D may inhibit *lncBCAS1-4_1* (*BCAS1-4_1*), causing inhibition of epithelial/mesenchymal transition (EMT), an important effect in invasion and metastasis.

**Table 1 ijms-23-03189-t001:** Long noncoding RNAs that were documented to mediate the effect of vitamin D3 in breast cancer through their interaction with vitamin D receptor.

LncRNA	Gene (Full Name)	Mechanism	Role in Cancer Transformation	References
*AFAP1-AS1*	AFAP1 antisense RNA 1	Induction of cell proliferation and EMT ^(a)^	Oncogene	[64]
*MALAT1*	Metastasis-associated lung adenocarcinoma transcript 1	Interaction with the TEAD (TEA domain) family member to suppress breast cancer cell migration, invasion, and metastasis	Tumor suppressor	[21,65]
*LINC00511*	Long intergenic non-protein-coding RNA 511	Sponging *miR-185-3p* and activating NANOG transcription	Oncogene	[21,66]
*LINC00346*	Long intergenic non-protein-coding RNA 346	Promotes cell proliferation and suppresses apoptosis	Oncogene	[21,67]
*SNHG6*	Small nucleolar RNA host Gene 6	Promotes EMT, cell proliferation, and migration through regulation of *miR-26a-5p*/MAPK6 signaling	Oncogene	[68,69,70]
*SNHG16*	Small nucleolar RNA host Gene 16	Sponging *miR-16-5p* and other miRNAs, promotes invasion and metastasis	Oncogene	[21,71]

^(a).^ Abbreviations: EMT, epithelial/mesenchymal transition; MAPK6, mitogen-activated protein kinase 6; TEAD, TEA domain family member.

**Table 2 ijms-23-03189-t002:** Long noncoding RNAs that are putative candidates to mediate the effect of vitamin D3 in breast cancer. Some mechanisms underlying such possible effects are also presented. Abbreviations are defined in the main text.

LncRNA	Gene (Full Name)	Mechanism	Role in Cancer Transformation	References
*GATA3-AS1*	GATA3 antisense RNA 1	Regulation of GATA3 expression, which is dependent on vitamin D/VDR signaling		[73,74,75,76]
*H19*	H19 imprinted maternally expressed transcript	Maintenance of the right proportion of oncogenic to tumor-suppressing lncRNAs; EMT through interaction with p53 and TNFAI8	Oncogene	[81,82,83,84]
*HOTAIR*	HOX transcript antisense RNA	Promotes the proliferation, migration, and invasion of breast cancer by regulating the *miR-601/ZEB1* axis	Oncogene	[98,99,100,101]
*HOTTIP*	HOXA distal transcript antisense RNA	Regulation of the right proportion between oncogenic and tumor-suppressing lncRNAs, promoting migration, invasiveness, and EMT by regulating the Wnt/β-catenin pathway; facilitates the stemness of breast cancer cells through upregulation of OCT4 and SOX2 by regulating *miR-148a-3p*/Wnt1 signaling	Oncogene	[85,86]
*Kcnq1ot1*	KCNQ1 opposite strand/antisense transcript 1	CCNE2 modulation by sponging *miR-145* and *hsa-miR-107*	Tumor suppressor	[88,89,90]
*lincRNA-p21*	Alias *TRP53COR1* (P53 pathway corepressor 1 protein tumor)	Induced by chemotherapy and mediated by a mutated p53; expressed in tumor-associated macrophages and prevents MDM2 from antagonizing p53 activation	Oncogene	[91,92]
*MEG3*	Maternally expressed gene 3	Inhibition of proliferation, invasion, and angiogenesis by downregulation of Akt signaling; EMT through Wnt/β-catenin signaling	Tumor suppressor	[93,94,95]
*Nespas*	GNAS antisense RNA 1	Reduces expression of encoding *miR-296-5p*, which targets telomerase in breast cancer cells and reduces telomerase activity, drives telomere shortening, and causes proliferation defects by enhancing senescence and apoptosis	Oncogene	[87]
*SNHG12*	Small nucleolar RNA host gene 12	Sponging *miRNA-675*	Tumor suppressor	[77,78,79]
*TOPORS-AS1*	TOPORS antisense RNA 1	Regulation of cell proliferation and apoptosis through p53/TP53 stability via ubiquitin-dependent degradation	Tumor suppressor (?)	[102,103]

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
