# Peer review of "Vitamin D May Protect against Breast Cancer through the Regulation of Long Noncoding RNAs by VDR Signaling"

_ijms, 2022, doi:10.3390/ijms23063189_

Round 1

Reviewer 1 Report

  1. The authors should edit the final paragraph of the conclusion, where they discuss the current limitations of investigating the role of lncRNAs in breast cancer. The current statements make it sound like it is a fruitless line of investigation. Please change the overall impression of those statements, and make it sound constructively critical. Expand on possible solutions, and methods to gain more confidence and specificity in associating lncRNAs to VD3 signaling, and their role in BC.
  2. For some of the lncRNAs with reported role in VD3 signaling in other cancers, the authors should include reports of their expression in BC, is information is available. That would help suggest more functional relevance of those lncRNAs in regulating VD3 signaling in BC.
  3. Cross-check the language of the manuscript, and make suitable edits. Typographical errors like 'BS' in page 5, line 157, 'breast cancer' instead of BC in page 7, line 262 remain.

Author Response

Comment: 1. The authors should edit the final paragraph of the conclusion, where they discuss the current limitations of investigating the role of lncRNAs in breast cancer. The current statements make it sound like it is a fruitless line of investigation. Please change the overall impression of those statements, and make it sound constructively critical. Expand on possible solutions, and methods to gain more confidence and specificity in associating lncRNAs to VD3 signaling, and their role in BC.

Answer: We wanted to present the main challenges associated with studies on the involvement of lncRNAs in the effect of vitamin D on breast cancer. To add some constructive elements, we have changed the last paragraph in the concluding section from:

“Exploring the role of lncRNAs in vitamin D effects in BC may be especially difficult due to (1) heterogeneity of BC cases, (2) pleiotropic biological effects of vitamin D, and (3) a great number of genes regulated by almost every, if not every, single lncRNA [51]. If we add to these points specific features of an individual and variable environmental conditions, the picture of the mutual relationships between BC, VD3 and lncRNAs seems to be quite complex. Therefore, high-throughput technologies and -omics attitude may be of a limited value in addressing this subject as compared with “traditional” approach, considering as many specific aspects as possible. Despite these objections, we argue that lncRNAs may contribute to the effects of vitamin D in BC, but at present it cannot be ad hoc determined whether such effects could be beneficial or harmful as they depend on many factors, including the cancer-related nature of a lncRNA – oncogenic or tumor sup-pressing.”

into:

“Exploring the role of lncRNAs in vitamin D3 effects in BC may be especially difficult due to (1) heterogeneity of BC cases, (2) pleiotropic biological effects of vitamin D3, and (3) a great number of genes regulated by almost all, if not all, lncRNAs [51]. Therefore, a special attention should be paid on the determination of specific breast cancer cases in clinical research and homogeneity of breast cancer cell lines in laboratory studies. This might limit the problem (2) with the different action of vitamin D3 in different breast cancer types. It may be expected that vitamin D3 may be effective in some specific breast cancer types, whereas its efficiency in others can be low, if any, and analysis of lncRNAs involved in vitamin D3 action may be helpful to anticipate the output of such action. Results of studies on cancer-related nature of lncRNAs – oncogenic or tumor suppressing may also help to foresee the effect of vitamin D3 as either beneficial or harmful.”  

Comments: For some of the lncRNAs with reported role in VD3 signaling in other cancers, the authors should include reports of their expression in BC, is information is available. That would help suggest more functional relevance of those lncRNAs in regulating VD3 signaling in BC.

Answer: It has been done in the original manuscript. Please compare the sections “3. Long Non-Coding RNAs in the Vitamin D Signaling in Cancer” and “4. Long Non-Coding RNAs and Vitamin D Signaling in Breast Cancer” and Table 1 and Table 2.

Comment: Cross-check the language of the manuscript, and make suitable edits. Typographical errors like 'BS' in page 5, line 157, 'breast cancer' instead of BC in page 7, line 262 remain.

Answer: We have done our best to correct and improve the language and the style of the manuscript.

Reviewer 2 Report

Major comments:

  1. The use of reference needs to be improved. Please use the most appropriate reference and not just any that may be found in Pubmed. Thus, please critically review all used references.
  2. The topic ncRNA needs a general introduction about the impact in gene regulation.
  3. The use of figures would be more illustrative than tables.
  4. The main role of vitamin D in breast cancer is its prevention and not its treatment. Please address this appropriately.

Minor comments:

  1. Please do not use the abbreviation "VD3", it is not common in the field. Better always write the term out.
  2. All gene and RNA abbreviations should be in italic, this applies also to ncRNAs.
  3. Please follow lates nomenclature (see, e.g., GeneCards). Please check the whole manuscript including figures.
  4. Also the protein names should be checked, please use latest international standards and a consistent nomenclature.
  5. All abbreviations should be explained at first time use and should then be applied consistently. This applies also to gene names in the Abstract.
  6. Better write "BC" out, it increases the readability of the manuscript.

Author Response

Comment: The use of reference needs to be improved. Please use the most appropriate reference and not just any that may be found in Pubmed. Thus, please critically review all used references.

Answer: It is difficult to address this remark as it does not specify an inappropriateness in our citations and reference list. We keep on maintaining our reference list in the original manuscript.

Comment: The topic ncRNA needs a general introduction about the impact in gene regulation.

Answer: “Non-coding RNAs” include many types of RNAs with functions of some of them that are poorly known. They are microRNA (miRNA), small interfering RNAs (siRNAs), piwi-interacting RNAs (piRNAs), promoter-associated transcripts (PATs), enhancer RNAs (eRNAs), circular RNAs (circRNAs), tRNA, snoRNA, lncRNAs and others. That is why we consider description of the role of all these ncRNAs in the regulation of gene expression to be done in a reliable way would require addition of a quiet large fragment to the Introduction section. That is why we have limited to the following fragment on basic mechanisms of the regulation of gene expression by lncRNAs, which we have added to the Introduction section (basic, textbook-like information that does not be referenced):

“Basic functions of lncRNAs in the regulation of gene expression can be arbitrary divided into: antisense/signaling, guide, scaffold, and decoy. A lncRNA may pair with a complementary fragment of mRNA, preventing or inhibiting its translation. Transcription can be influenced by a lncRNA through recruitment and/or guiding transcriptional activators and repressors to the target gene. A lncRNA may serve as a platform (scaffold) to facilitate assembling a chromatin remodeling complex to alternate the structure of chromatin to more permissive or repressive for transcription factors. A lncRNA may act as a decoy to recruit other regulatory RNAs or transcription factors and sequester them from their target mRNA or DNA. “A sponging effect” occurs when a single lncRNA binds many miRNAs, preventing interaction with their targets.”

Comment: The use of figures would be more illustrative than tables.

Answer: Presenting all information from the tables in a single figure would produce so complex picture that its description (caption) would be so long and complex that would likely obstruct readability of such figure. On the other hand, presenting a separate figure for each of the proteins mentioned in Table 1 and 2, would result in so many figures, enough complex each that the manuscript would be rather a poster than a paper.  

Comment: The main role of vitamin D in breast cancer is its prevention and not its treatment. Please address this appropriately.

Answer: We have added the following sentence to the concluding section:

“However, it should be stressed that vitamin D is considered in much greater extent as a preventive than a therapeutic compound.”

Comment: Please do not use the abbreviation "VD3", it is not common in the field. Better always write the term out.

Answer: We have followed this suggestion throughout the revised manuscript.

Comment: All gene and RNA abbreviations should be in italic, this applies also to ncRNAs.

Answer: We have followed this suggestion throughout the revised manuscript.

Comment: Please follow lates nomenclature (see, e.g., GeneCards). Please check the whole manuscript including figures.

Answer: We have followed this suggestion throughout the revised manuscript.

Comment: Also the protein names should be checked, please use latest international standards and a consistent nomenclature.

Answer: We have followed this suggestion throughout the revised manuscript.

Comment: All abbreviations should be explained at first time use and should then be applied consistently. This applies also to gene names in the Abstract.

Answer: We have followed this suggestion throughout the revised manuscript with the exception of the Figure 2 legends and the Table 2 title that refers the reader to the main text as defining all abbreviations there would obstruct the readability of the legend/title.

Round 2

Reviewer 2 Report

Please verify, if you really mean vitamin D3 (and not 25(OHD3 or 1,25(OH)2D3), where you now changed VD3 to vitamin D3. 

Author Response

Comment: Please verify, if you really mean vitamin D3 (and not 25(OHD3 or 1,25(OH)2D3), where you now changed VD3 to vitamin D3.

Answer: In this review, the term „vitamin D3” is used in a broad sense: in in vivo studies either chemically pure vitamin D3 or its modifications are used and the concentration of 25(OH)D in serum is determined. In in vitro studies, typically lacking vitamin D3-metabolizing system, 1,25(OH)2D is used.  That is why generalizing the 1,25(OH)2D-related results to vitamin D3 might be chemically incorrect, but biologically usually justified. For a concise review, we use “vitamin D3” in a general sense, as the term vitamin D receptor is commonly used, although it is the 1,25(OH)2D receptor.